# Co/SiO_2_ Catalyst for Methoxycarbonylation of Acetylene: On Catalytic Performance and Active Species

**DOI:** 10.3390/molecules29091987

**Published:** 2024-04-26

**Authors:** An Wang, Hongchen Cao, Leilei Zhang, Aiqin Wang

**Affiliations:** 1CAS Key Laboratory of Science and Technology on Applied Catalysis, iChEM, Dalian Institute of Chemical Physics, Chinese Academy of Sciences, Dalian 116023, China; wangan@dicp.ac.cn (A.W.); hongchencao@dicp.ac.cn (H.C.); zhangleilei@dicp.ac.cn (L.Z.); 2University of Chinese Academy of Sciences, Beijing 100049, China

**Keywords:** methoxycarbonylation of acetylene, noble metal-free, cobalt carbonyls, Co leaching

## Abstract

Reppe carbonylation of acetylene is an atom-economic and non-petroleum approach to synthesize acrylic acid and acrylate esters, which are key intermediates in the textile, leather finishing, and polymer industries. In the present work, a noble metal-free Co@SiO_2_ catalyst was prepared and evaluated in the methoxycarbonylation reaction of acetylene. It was discovered that pretreatment of the catalyst by different reductants (i.e., C_2_H_2_, CO, H_2_, and syngas) greatly improved the catalytic activity, of which Co/SiO_2_-H_2_ demonstrated the best performance under conditions of 160 °C, 0.05 MPa C_2_H_2_, 4 MPa CO, and 1 h, affording a production rate of 4.38 g_MA+MP_ g_cat_^−1^ h^−1^ for methyl acrylate (MA) and methyl propionate (MP) and 0.91 g_DMS_ g_cat_^−1^ h^−1^ for dimethyl succinate (DMS), respectively. Transmission electron microscopy (TEM), X-ray diffraction (XRD), and diffuse reflectance infrared Fourier transform spectra of CO adsorption (CO-DRIFTS) measurements revealed that an H_2_ reduction decreased the size of the Co nanoparticles and promoted the formation of hollow architectures, leading to an increase in the metal surface area and CO adsorption on the catalyst. The hot filtration experiment confirmed that Co_2_(CO)_8_ was generated in situ during the reaction or at the pre-activation stage, which served as the genuine active species. Our work provides a facile and convenient approach to the in situ synthetization of Co_2_(CO)_8_ for a Reppe carbonylation reaction.

## 1. Introduction

Reppe carbonylation reactions of acetylene (including hydrocarboxylation, alkoxycarbonylation, etc.) (Figure 1) provide atom-economic and non-petroleum routes to produce acrylic acid (AA), methyl acrylate, and dimethyl succinate [1,2], which are raw materials and key intermediates that are widely used in the textile, leather finishing, and polymer industries, with a global demand of 10 million tons annually [3,4,5].

Various catalysts have been developed for the carbonylation reactions of acetylene, of which Ni-based catalysts are the original ones [6]. Reppe and coworkers were the first to discover that Ni(CO)_4_ was able to catalyze the hydrocarboxylation of acetylene to produce AA under 3 MPa CO and acetylene (1/1, *v*/*v*) [2]. Following this, other nickel catalysts, such as (copper-promoted) nickel halides [7,8], nickel acetates [9], and Ni-P complexes [10], were also reported. Among others, Badische Anilin-und-Soda-Fabrik (BASF) developed the commercial catalytic system NiBr_2_-CuBr_2_-CH_3_SOOOH for hydrocarbonylation of acetylene [11], which, however, suffers from a difficulty in the separation of catalyst products [12]. To tackle this problem, metal oxides (such as SiO_2_, Al_2_O_3_, vermiculite, and MCM-41) [13,14,15] and zeolites (e.g., NaY and ZSM-5) [16] were applied as support materials of Ni. However, these catalysts have one or more drawbacks, which include a short catalyst lifetime, harsh reaction conditions, and metal leaching. For example, Shi and coworkers reported that Ni/Y served as an efficient catalyst for the hydrocarboxylation of acetylene [17], and the AA space-time yield reached 62 g_acrylic acid_ g_cat_^−1^ h^−1^ under reaction conditions of 235 °C and 3.6 MPa. Although they claimed that the catalysis followed a heterogeneous pathway, the Ni/Y material suffered from severe coking. Yan and coworkers prepared a NiO/AlOOH catalyst, which gave an AA space-time yield as high as 412 g_acrylic acid_ g_cat_^−1^ h^−1^, yet it was discovered that the leached Ni-carbonyls were the true active species [12]. Given the high toxicity and instability of Ni(CO)_4_, which might generate in situ during a reaction, the potential application of Ni-based catalysts is limited.

Other catalysts based on group VIII transition metals, such as Pd, Ir, and Ru, have also been explored [18]. For instance, Drent and coworkers developed a Pd complex with the ligand of 2-pyridylphosphine (2-PyPPh_2_) [19], which exhibited excellent catalytic performance in the methoxycarbonylation of propyne, and the TOF reached as high as 40,000 h^−1^ with 99.95% selectivity to methyl methacrylate. The 2-pyridyl ring was found to be crucial for the catalytic activity, which, if replaced by a phenyl or 4-pyridyl group, led to a drastic decrease in TOF. Nonetheless, strong acid promoters, e.g., p-toluenesulfonic acid, are indispensable for the reaction, which would cause serious environmental issues [20]. Ding and coworkers developed porous organic polymer-supported single-site Pd catalysts, in which the 2-pyridylphosphine ligand and p-toluenesulfonic acid were incorporated into the framework of the support [21,22], and the methoxycarbonylation reaction of acetylene was allowed to run in the absence of an extra acid promoter. However, this approach suffered from tedious procedures for the preparation of the support [23,24]. In addition, the high cost and low abundance of Pd compromised its efficiency.

Cobalt is also the metal of choice for Reppe carbonylation reactions due to its earth abundance and high catalytic activity [25,26,27]. Pyridine-promoted cobalt carbonyls demonstrated high efficiency in converting butadiene to adipic acid in a pilot-scale process developed by BASF [28,29]. Cobalt carbonyls were also competent catalysts in the alkoxycarbonylation reactions of alkenes [30]. Nevertheless, to the best of our knowledge, few reports have been produced on the Co-catalyzed methoxycarbonylation of acetylene. In the present work, we prepared SiO_2_-supported Co catalysts and investigated their catalytic performance in the methoxycarbonylation of acetylene in the absence of acid promoters. It was discovered that the pre-activation atmosphere (i.e., H_2_, C_2_H_2_, syngas, CO, and NH_3_) had a profound impact on the catalytic activity, which increased following the trend of C_2_H_2_ < CO < syngas ≈ H_2_. Under reaction conditions of 160 °C and 0.05 MPa C_2_H_2_ and 4 MPa CO, Co/SiO_2_ exhibited high catalytic activity, leading to a production rate of 4.2 g_MA+MP_ g_cat_^−1^ h^−1^ for MA and MP and 0. 9 g_DMS_ g_cat_^−1^ h^−1^ for DMS, respectively. Mechanism studies by XRD and hot filtration test revealed that Co nanoparticles underwent dynamic evolution during the reaction, where cobalt carbonyls were formed in situ and served as the genuine active species. Our work provided an approach for the in situ generation of catalytically active species for the methoxycarbonylation reaction, thus circumventing the employment of costly and unstable conventional Co_2_(CO)_8_ catalysts, which are generally synthesized under harsh conditions of high temperatures and high pressures [31,32,33].

## 2. Results and Discussion

The Co/SiO_2_ catalysts were prepared by the one-pot synthetic method, followed by calcination and reduction [34]. The XRD patterns (Figure 1a) exhibited characteristic diffraction peaks of metallic Co with an fcc structure (JCPDS 00-01-1259), and the size of the Co was calculated to be 28 nm according to the Scherrer equation [35]. No diffraction peaks ascribed to SiO_2_ (JCPDS 00-33-1161) were present, probably owing to the amorphous nature of SiO_2_ in the sample. The TEM images (Figure 1b) and energy-dispersive X-ray (EDX) spectra (Figure 1c) showed that Co nanoparticles were homogeneously distributed on SiO_2_, most of which had an average size of 18.6 nm. However, the examination of different regions showed that there were also smaller Co nanoparticles with sizes of 1–2 nm (Appendix A). The lattice spacing was measured to be 2.07 nm and 1.93 nm (Figure 1d), corresponding to the (111) and (101) plane in fcc Co, respectively, which were in agreement with the XRD measurements.

The as-prepared Co/SiO_2_ catalysts were evaluated in the methoxycarbonylation reaction of acetylene under conditions of 0.05 MPa C_2_H_2_, 4 MPa CO, 5 mL CH_3_OH, and 160 °C. However, no products were detected in the three catalysts with different Co loadings after a 1 h reaction (Table 1, entries 1–3). It is assumed that the passive layer of CoO_x_ outside of the metallic Co might suppresses the reactant to access the catalytically active sites [36]; therefore, 1 MPa H_2_ was charged into the reactor to activate the catalyst in situ. As expected, under this condition, the Co/SiO_2_ samples demonstrated high catalytic performance for the target reaction. Three main products were detected (i.e., MA, MP, and DMS) after a 1 h reaction (Table 1, entries 4–6), of which MA and DMS were produced by the methoxycarbonylation and dimethoxycarbonylation of acetylene, respectively, while MP was derived from the hydrogenation of MA. Specifically, in the 41.7% Co/SiO_2_ catalyst (Table 1, entry 4), all the MA had been completely hydrogenated to MP, which was produced with a rate as high as 4.42g_MP_ g_cat_^−1^ h^−1^, and DMS was yielded with a relatively lower rate of 0. 58 g_DMS_ g_cat_^−1^ h^−1^. The other two Co/SiO_2_ catalysts with different Co loadings were also tested in the reaction. The 31.6% Co/SiO_2_ sample showed similar catalytic performance to the 41.7% Co/SiO_2_ sample (Table 1, entry 5). However, the catalytic activity of the 25.6% Co/SiO_2_ sample, with an even lower Co loading, decreased drastically (Table 1, entry 6), and only the methoxycarbonylation product was yielded, with a rate of 0.39 g_MA+MP_ g_cat_^−1^ h^−1^. The reason might be that with a decrease in Co loading, the size of the Co became smaller, and in turn, the interaction between Co and SiO_2_ became stronger (e.g., cobalt silicate might be formed) [34,37], leading to a more difficult reduction of the CoO_x_ species. The other support-material-loaded Co catalysts were also evaluated in the reaction. However, Co/WO_x_, Co/HAP, Co/TiO_2_, and Co/S-1 were totally inactive for the reaction (Table 1, entries 7–10). Metallic Co powder was inefficient for the reaction as well (Table 1, entry 11).

As the pre-activation of Co/SiO_2_ by H_2_ improved the catalytic activity remarkably, the other reductants, i.e., C_2_H_2_, CO, syngas, and NH_3_, were also employed to pre-activate the sample to see if the catalytic activity could be further improved. Except for NH_3_, the pre-activation was conducted in an autoclave in CH_3_OH under different reductant gases at 160 °C for 1 h, and after that, the autoclave was flushed with N_2_ and recharged with the reactant gas (0.05 MPa C_2_H_2_ and 4 MPa CO), and the reaction was allowed to run at 160 °C for 1 h. The pretreatment by NH_3_ was carried out according to the literature [38]. As shown in Table 2, the pretreatment under Ar made no contribution to the activity (Table 2, entry 1), thus precluding the possibility that CH_3_OH served as the reductant. By contrast, the pre-activations under C_2_H_2_, CO, or syngas atmosphere all led to obvious yields of the three products, and the production rate increased following the trend of Co/SiO_2_-C_2_H_2_ (Table 2, entry 2) < Co/SiO_2_-CO (entry 3) < Co/SiO_2_-syngas (entry 5) ≈ Co/SiO_2_-H_2_ (entry 4). The improvement in catalytic activity by pretreatment with C_2_H_2_ or CO was surprising, as no products were detected using the as-made Co/SiO_2_ catalyst under 0.05 MPa C_2_H_2_ and 4 MPa CO at 160 °C for 1 h, as mentioned before (Table 1, entry 1). We suppose there might be an induction period of >1 h; accordingly, the reaction was once again conducted using an as-made Co/SiO_2_ catalyst under 0.05 MPa C_2_H_2_ and 4 MPa CO for a long time (i.e., 2 h); however, no products were detected. On the contrary, the Co/SiO_2_-NH_3_ sample showed no catalytic activity for the reaction (Table 2, entry 6). The reason for this might be that NH_3_ adsorbed strongly on the surface of metallic Co, and thereby, the adsorption of the reactants on the catalyst was prevented. To verify this, we carried out a control experiment, where the methoxycarbonylation of acetylene was conducted under conditions of 0.1 MPa NH_3_, 0.05 MPa C_2_H_2_, 4 MPa CO, 5 mL CH_3_OH, 160 °C, and 1 h using Co_2_(CO)_8_ as the catalyst (Co_2_(CO)_8_ was used here because it has been proven to be the genuine active species in the following section). It was found that no product was yielded, suggesting that the Co_2_(CO)_8_ catalyst was poisoned by NH_3_.

On the basis of the above results, it is inferred that pretreatment with a distinct reducing gas should modify the structure of the Co/SiO_2_ catalyst, where the active species are more prone to be formed. Accordingly, TEM, XRD, and CO-DRIFT measurements of the activated catalysts were conducted. Figure 2 shows the TEM images and particle size histograms for different pre-activated Co/SiO_2_ samples. The average particle size of metallic Co was 24.9 nm, and it was 20.3 nm for Co/SiO_2_-C_2_H_2_ (Figure 2a,d) and Co/SiO_2_-CO (Figure 2b,e), respectively. Compared with fresh Co/SiO_2_ (18.6 nm), it is obvious that the C_2_H_2_ and CO pre-activation induce an increase in particle size. In sharp contrast, in the Co/SiO_2_-H_2_ sample (Figure 2c,f and Appendix A), not only had the size of the Co decreased to 17.5 nm, but also, a lot of Co nanoparticles had been reconstructed to hollow shells. This reconstruction during H_2_ reduction might result from the strain induced by a variation in the metal–metal distances between the CoO_x_ surface layer and the inner metallic Co [39].

As shown in Figure 3a, the intensity of the diffraction peaks at 44.4° and 47.3°, attributed to the (111) and (101) plane of metallic Co [40] in the samples of Co/SiO_2_-C_2_H_2_ and Co/SiO_2_-CO, increased greatly compared with those of the fresh Co/SiO_2_, indicating an increase in the size of the Co nanoparticles. In sharp contrast, for the Co/SiO_2_-H_2_ sample, the diffraction peaks attenuated instead, suggesting a decrease in the Co particle size, or a reduction in the crystallinity of the metallic Co. These results were consistent with the TEM examination results but quite different from conventional experimental results, where H_2_ reduction generally results in the sintering of metal nanoparticles [41]. CO-DRIFTs were also carried out to study the electronic properties of Co. As shown in Figure 3b, four adsorption bands appeared on different samples. The band at 2048 cm^−1^ was ascribed to the ν(C≡O) of CO molecules that were linearly adsorbed on metallic Co [42], while the one at 1793 cm^−1^ was assigned to the ν(C≡O) of CO molecules in multi-bonded carbonyls on Co [43]. The other two bands at 1627 and 1510 cm^−1^ were attributed to the ν_asymm_(C=O) of bicarbonate species and ν_asymm_(C=O) of carbonate species, respectively [44,45]. It was found that the integrated area of the CO band at 2048 cm^−1^ and 1793 cm^−1^ increased with the trend of Co/SiO_2_-C_2_H_2_ < Co/SiO_2_-CO < Co/SiO_2_-H_2_, indicating an increased surface area of the metallic Co, which should come from the reduction of the passive layer of CoO_x_. In addition, this peak red-shifted to 2030 cm^−1^ on Co/SiO_2_-CO and further to 2019 cm^−1^ on Co/SiO_2_-H_2_, which, according to the TEM images, might result from the increased proportion of corner and edge sites in the Co nanoparticles with a decrease in sizes.

Stability is a key criterion for judging the quality of a catalyst [46]. Therefore, we also studied the recyclability of the Co/SiO_2_ catalyst. When the used catalyst was separated from the reaction mixture by filtration, it was found that the filtrate was dark red in color, suggesting severe leaching of Co [47,48]. The concentration of Co in the filtrate of different samples increased, following the trend of Co/SiO_2_-C_2_H_2_ < Co/SiO_2_-CO < Co/SiO_2_-H_2_ (Figure 3c), which was in agreement with that of the catalytic activity. In addition, the Co/SiO_2_ catalyst was discovered to already undergo Co leaching after the pretreatment with the reductant. In particular, for the Co/SiO_2_–syngas sample, after the pre-activation using syngas, the solid sample was separated by filtration and was then subjected to a batch of reactions (Appendix A); however, no products were yielded at all. By contrast, when the dark-red filtrate (containing 200 ppm Co) was employed as a catalyst, production rates of 3.76 g_MA+MP_ g_cat_^−1^ h^−1^ of MA and MP and 0.49 g_DMS_ g_cat_^−1^ h^−1^ of DMS were achieved (Table 3), which was quite similar to that of the Co/SiO_2_–syngas (Table 2, entry 5). These results indicated that it was the leached Co species that contributed to the overall catalytic activity. ^13^C NMR measurement was then conducted to identify the leached Co species; however, the filtrate did not show any signal because of the paramagnetic nature of Co. The yellow color of the filtrate implied that the leached Co species might be Co_2_(CO)_8_. To confirm this point, we used commercial Co_2_(CO)_8_ as a catalyst for the methoxycarbonylation of acetylene. It was discovered that Co_2_(CO)_8_ demonstrated similar catalytic activity and product distribution to the filtrate (Figure 4), verifying that Co_2_(CO)_8_ was the true active species for the reaction.

Co_2_(CO)_8_ is known as an efficient catalyst for the alkoxycarbonylation and hydroformylation of alkenes [49,50,51], although no reports on its catalytic performance in the methoxycarbonylation of acetylene have been published. Our work thus demonstrated that Co_2_(CO)_8_ is also a competent catalyst for this transformation to produce MA and MP. Traditionally, Co_2_(CO)_8_ is quite unstable, and its synthesis requires either harsh reaction conditions (i.e., 140–230 °C, 10–70 MPa) or stoichiometric reducing reagents (such as metal powders, NaBH_4_, or Na_2_S_2_SO_3_) or phosphine oxide promoters. Li and coworkers recently reported that Co_2_(CO)_8_ could be formed in situ from Co/MoS_2_ under conditions of 6 MPa CO, 140 °C, and 15 h, and MoS_2_ was claimed to be critical for this process [28]. This work demonstrated that when using conventional SiO_2_ as a support material, Co_2_(CO)_8_ could also be generated in situ under conditions of 4 MPa CO, 1 MPa H_2_, 160 °C, and 1 h, thus providing a facile, straightforward approach to prepare Co_2_(CO)_8_ for Reppe carbonylation reactions.

## 3. Materials and Methods

The Co/SiO_2_ was prepared according to the literature [34]. Briefly, 2.91 g Co(NO_3_)_2_·6H_2_O (10 mmol) and 2.08 g TEOS (10 mmol) were dissolved in 50 mL mixed liquor of water and ethanol (3/1, *v*/*v*) and stirred for 10 min. After, 5 mL of NH_3_·H_2_O was added to the above solution, and the suspension was stirred at room temperature for another 8 h. The precipitate was separated and collected by filtration, washed with deionized water, and dried at 100 °C overnight. The obtained solid was calcined at 500 °C in a muffle furnace for 4 h and was then reduced in flowing pure hydrogen (100 mL/min) for 3 h at 600 °C. The obtained Co/SiO_2_ catalyst was labeled as 41.7% Co/SiO_2_. The other two catalysts with different Co loadings (i.e., 31.6% Co/SiO_2_ and 25.6% Co/SiO_2_) were prepared using similar procedures, except for changing the amount of TEOS to 15 and 20 mmol, respectively.

The other support material (i.e., WO_x_, hydroxyapatite (HAP), and TiO_2_)-loaded Co catalysts were synthesized by the impregnation method. First, 5 mL Co(NO_3_)_2_·6H_2_O solution (10 mg _Co_/mL) and 5 mL ultrapure water were mixed and stirred to form a transparent solution, and then, 1 g carrier was added to the above solution. After the mixture was stirred at room temperature for 24 h, the excess water was removed by rotary evaporation until dry. The resulting power was further dried at 373 K overnight and then reduced in flowing pure hydrogen (100 mL/min) for 3 h at 600 °C.

The preparation of the Co_2_(CO)_8_ solution in methanol from Co/SiO_2_ started with adding 10 mg of fresh Co/SiO_2_ to a 30 mL autoclave with 5 mL methanol, and then, the autoclave was purged with N_2_ repeatedly (4 times) to remove gaseous and dissolved oxygen. After that, the autoclave was charged with 4 MPa CO and 1 MPa H_2_, and the mixture was stirred (600 rpm) at 160 °C for 4 h. Then, the autoclave was placed into an ice bath until the temperature was below 5 °C. The solid was separated by filtration, and the yellow filter liquor was collected and kept at 0 C.

Other comparison materials, including Co power and Co_2_(CO)_8_, are commercial samples at analytically pure levels.

### 3.1. Characterization

The concentration of Co in the filtrate after pretreatment or reaction was determined by inductively coupled plasma spectroscopy (ICP-AES) on an IRIS Intrepid II XSP instrument (Thermo Electron Corporation, Waltham, MA, USA). The patterns of XRD were recorded on a PANalytical X’ pert diffractometer with a Cu-Kα radiation source, operated at 40 kV and 40 mA under a continuous mode in the 2θ range of 10°~90°. The morphology and element distribution of samples were observed by STEM and EDS experiments, which were performed on a JEM-2100F Transmission Electron Microscope (JEOL, Singapore) with a spatial resolution of 0.19 nm at 20 kV, equipped with the Oxford Instruments ISIS/INCA EDS system with an Oxford Pentafet Ultrathin Window (UTW) detector. Before the microscopy examination, the sample was demagnetization-treated using a magnet. After that, the sample was ultrasonically dispersed in ethanol for 5–10 min, and then, a drop of the suspension was dropped on a copper TEM grid coated with a thin holey carbon film. CO-DRIFTS (diffuse reflectance infrared Fourier transform spectroscopy) experiments were performed by means of a Bruker Equinox70 spectrometer (Bruker, Singapore), equipped with a mercury–cadmium–telluride detector at a resolution of 4 cm^−1^, using 16 scans in a range of 400–4000 cm^−1^. Prior to the measurement, the catalysts were pretreated in situ with H_2_ (30 mL/min) at 160 °C for 30 min, and then, the flow was switched to He (30 mL/min) at 170 °C for another 30 min to remove surface H. After that, the catalyst cooled down to room temperature, and background spectra were recorded at 20 °C in He flow.

### 3.2. Catalyst Evaluation

The acetylene methoxycarbonylation (AMC) reaction was performed in a 30 mL autoclave equipped with a quartz lining. The autoclave was filled with 10 mg of a catalyst and 5 mL methanol, and then, the autoclave was purged with N_2_ repeatedly (4 times) to remove gaseous and dissolved oxygen. After that, it was charged with 1 MPa 5% C_2_H_2_-95% He, 1 MPa H_2_, and 4 MPa CO. The reaction was started by heating the mixture to 160 °C under vigorous stirring at a speed of 600 rpm. After the reaction, the reactor was cooled to room temperature, and the liquid product was analyzed using n-pentanol as the internal by gas chromatography (Agilent 7890B, Santa Clara, CA, USA), equipped with an HP-INNO WAX column (30 m × 320 μm × 0.25 μm). Only methyl acrylate (MA), methyl propionate (MP), dimethyl succinate (DMS), 1,1-dimethoxypropane (2,2-DMP), and 3-pentanone were detected in the liquid product. The reaction selectivity was calculated based on the molar ratio of MA, MP, and DMS to the total liquid product, and the reaction rate was calculated by the total mass of MA, MP, and DMS produced per mass of Co per hour. The ‘others’ in the result are the total mass of 3-pentanone and 2,2-DMP. It should be emphasized when using Co_2_(CO)_8_ as a catalyst before the reactant gas is charged that the autoclave should be placed into an ice bath until the temperature is below 5 °C; otherwise, the dissolved Co_2_(CO)_8_ will be swept away owing the volatile property of Co_2_(CO)_8_.

The pretreatment of the Co/SiO_2_ catalyst and the catalytic evaluation were conducted in a methoxycarbonylation reaction of acetylene. Taking Co/SiO_2_-C_2_H_2_ as an example, 10 mg of fresh Co/SiO_2_ was added into a 30 mL autoclave with 5 mL methanol, and then, the autoclave was purged with N_2_ repeatedly (4 times) to remove the air and dissolved oxygen. After that, the autoclave was charged with 5% C_2_H_2_–95% He until 1 MPa. The autoclave was heated at 160 °C for 1 h with vigorous stirring at a speed of 600 rpm. When the reactor was cooled to room temperature, the gas in the reactor was evacuated, and the autoclave was recharged with C_2_H_2_ (1 MPa) and CO (4 MPa) for acetylene methoxycarbonylation. A similar pretreatment process was also conducted using 1 MPa H_2_, 4 MPa CO, or syngas (a mixture of 1 MPa H_2_ and 4 MPa CO).

## 4. Conclusions

In summary, a noble metal-free Co@SiO_2_ catalyst was prepared and tested in the methoxycarbonylation reaction of acetylene. The pre-activation of the catalyst using different reductant gases (i.e., C_2_H_2_, CO, H_2_, and syngas) was found to be crucial for the high catalytic performance. The catalytic activity increased following the trend of Co/SiO_2_-C_2_H_2_ < Co/SiO_2_-CO < Co/SiO_2_–syngas ≈ Co/SiO_2_-H_2_, and high production rates of 4.38 g_MA+MP_ g_cat_^−1^ h^−1^ and 0.91 g_DMS_ g_cat_^−1^ h^−1^ were obtained for Co/SiO_2_-H_2_ under conditions of 160 °C, 0.05 MPa C_2_H_2_, 4 MPa CO, and 1 h. The characterization by TEM, XRD, and CO-DRIFTS revealed that the size or crystallinity of Co nanoparticles was greatly decreased upon pre-reduction by H_2_, which remarkably increased the metal surface area and CO adsorption on the catalyst. The hot filtration experiment and controlled measurement confirmed that Co was leached into the solution in the form of Co_2_(CO)_8_ and served as the genuine active species. Our work provides a facile and convenient approach to synthesize Co_2_(CO)_8_ for a Reppe carbonylation reaction, thus avoiding the direct usage of highly costly and unstable Co_2_(CO)_8_ as a catalyst.

## Data Availability

Data are contained within the article and Appendix A.

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
