# Peer review of "Co/SiO2 Catalyst for Methoxycarbonylation of Acetylene: On Catalytic Performance and Active Species"

_molecules, 2024, doi:10.3390/molecules29091987_

Round 1

Reviewer 1 Report

Comments and Suggestions for Authors

This is a nice study of acetylene methoxy carbonylation to methyl acrylate using Co/SiO2 catalysts.  The introduction is well-written, and the methodology is clear.  Results are also presented with care.  The study of the filtrate, in particular, is interesting.  

There are a few points I would like the authors to address. 

1. The DRIFT experiments show only the CO region.  The authors need to show the complete region and comment on them.  Are there carboxylates, carbonates, alkoxy, etc...? These might be important.

2. Another point is also related to the DRIFT experiment.  The authors indicated the following. 

[For the sample treated with different reductants, not only did the area of the broad CO-band significantly increase, but also they were red-shifted, suggesting an increase of surface area and electron density on Co nanoparticles.]

The authors are requested to explain, with references, why a red-shift of the nu CO is due to an increase in surface area.  Also, if the CO at 2048 cm-1 is attributed to metal CO, what is the meaning of an increase in electron density in the other spectra?  

[In particular, the maximum was reached on the Co/SiO2-H2 sample, suggesting the highest metallic surface area of the sample and thus the smallest size of Co, which was consistent with the TEM and XRD measurement results.]

Again, the authors make it sound that there is an understood relationship between the shift in the nu CO and metal surface area, without providing an explanation or previous work.

3. The highest yield is found for the smallest Co particles.  Since the authors have the TEM results for particle size distribution.  Can the authors compare the reaction rates per Co particle density too?

4. Would it be possible for the authors to conduct TPR and compute for the Co metal particles (granted only for the H2 reduced one) and compare the turnover frequency to that obtained for Co2(CO)8?

5. There is one metal, why co-precipitation?

[Co/SiO2 catalysts were prepared by the co-precipitation method followed by calcination and reduction.] 

Comments on the Quality of English Language

Here are a few examples of sentences requiring attention. 

When the spent catalyst was separated from the reaction mixture by filtration, it was found [that] the filtrate was dark red in color, ..

Here Our work...

Co nanoparticles were homogeneously distributed on the supported..

Reviewer 2 Report

Comments and Suggestions for Authors

In this work, the authors prepare a noble metal-free Co@SiO2 catalyst for methoxycarbonylation of acetylene. Some issues must be solved before acceptance.

1.     The full names of the abbreviations should be presented when they first appear. Necessary PDF cards should be provided.

2.     Please identify the SiO2 peak in the XRD pattern. Also, please well explain the role of SiO2 for the catalysis. Whether possible synergetic effects exist between Co and SiO2.

3.     This version lacks characterizations for material electronic structure. At least, Co and O XPS should be measured and fitted, especially for O 1s XPS. Because this reaction involves O species, the measurement and fitting of O 1s XPS is necessary. The authors should refer to this work 10.1039/D1TA10652J for this point.    

4.     Apart from the involved reductive gas, NH3 is also reductive and may be applied to improve the catalysis performance. In the perspective of this work, the authors can refer to this work 10.1063/5.0083059 to extend the possible usage of NH3 reductive gas.

Comments on the Quality of English Language

Minor editing of English language required

Round 2

Reviewer 1 Report

Comments and Suggestions for Authors

The authors have addressed the queries and modified the manuscript accordingly.  There is one misassignment in the IR section that needs to be corrected.  Formate species have asymmetric and symmetric stretches at about 1550 cm-1 and 1380 cm-1 (depending on the type of metal oxide).  The peak at 1510 cm-1 is not that of a formate species but of a carbonate species.  It is recommended that when reporting IR assignments to indicate the mode of absorption of a given species beside the frequency of that mode.  This usually prevents misassignments.
